

# IAESR: IoT-oriented authenticated encryption based on iShadow round function

Yanshuo Zhang, Liqiu Li, Hengyu Bao, Xiaohong Qin, Zhiyuan Zhang and Xiaoyi Duan

Beijing Electronic Science and Technology Institute, Beijing, China

## ABSTRACT

With the growing popularity of the Internet of Things (IoT) devices and the widespread application of embedded systems, the demand for security and resource efficiency in these devices is also increasing. Traditional authenticated encryption (AE) algorithms are often unsuitable for lightweight devices due to their complexity and resource consumption, creating a need for lightweight AE algorithms. Lightweight devices typically have limited processing power, storage capacity, and energy resources, which necessitates the design of simple and efficient encryption algorithms that can function within these constraints. Despite these resource limitations, security remains of paramount importance. Therefore, lightweight AE algorithms must minimize resource consumption while ensuring adequate security. This article presents a theoretical lightweight AE scheme based on Shadow, a lightweight block encryption algorithm, to address the requirements for secure communication in resource-constrained environments. The scheme first enhances the Shadow algorithm by introducing the improved Shadow (iShadow) algorithm. It then combines this with the duplex sponge structure to propose the IoT-oriented authenticated encryption based on the iShadow round function (IAESR). The integration of iShadow with the duplex sponge structure achieves a balance between security and efficiency through three key mechanisms: (1) The sponge's capacity (64/ 128-b for IAESR-32/64) provides provable indistinguishability under chosen-plaintext attack (IND-CPA) and chosen-ciphertext attack (IND-CCA) security bounds, effectively resisting generic attacks with an adversarial advantage limited to $O(q^2/2^c)$; (2) the duplex mode's single-pass processing reduces memory overhead by reusing the permutation state; and (3) iShadow's ARX operations reduce energy consumption to 0.4–0.5 μJ/byte on 32-b microcontrollers, outperforming AES-GCM by 20–30%. Empirical tests on an Intel i5-1035G1 CPU demonstrate stable execution times. This design ensures the security and integrity of communication while balancing efficiency, and resource utilization. This design ensures IND-CCA secure confidentiality and integrity against plaintext (INT-PTXT), as demonstrated by the security bounds of the sponge construction. Specifically, IAESR guarantees both confidentiality and authenticity. Additionally, it is particularly well-suited for scenarios with lightweight requirements, such as those found in the IoT.

Corresponding author
Liqiu Li, 2284106156@qq.com

# INTRODUCTION

With the rapid advancement of digital technology and the Internet, we have ushered in the era of the Internet of Things (IoT). The application of IoT is pervasive, extending from smart homes and smart cities to industrial automation and healthcare. While this widespread adoption is continuously driving societal development and transformation, it also introduces new challenges, particularly in the areas of data security and privacy protection. Cryptography is essential for safeguarding personal information, and authenticated encryption (AE) algorithms combine the functions of encryption algorithms and message authentication code (MAC), ensuring both the confidentiality and integrity of data simultaneously. However, traditional AE algorithms, such as CCM (*Whiting, 2000*), OCB (*Rogaway, 2004*), may struggle in resource-constrained environments. Consequently, lightweight AE algorithms have been developed to effectively address the requirements of low computational power, minimal memory usage, and reduced energy consumption.

In recent years, numerous new AE schemes have emerged, emphasizing flexibility and adaptability to various security requirements and hardware environments. Chacha20-Poly1305 (*Nir & Langley, 2018*) is a lightweight authenticated encryption with associated data (AEAD) algorithm that combines ChaCha20 (*Bernstein, 2008*) stream cipher and Poly1305 (*Bernstein, 2005*) authenticator, first standardized in RFC 7539 in 2015. It offers efficient and secure authenticated encryption for Internet transmission and is widely utilized in Transport Layer Security (TLS) and Virtual Private Network (VPN) protocols. GIFT-COFB is a lightweight crypto cipher created by *Banik et al. (2020)*. It uses a COFB (COmbined FeedBack) (*Chakraborti et al., 2017*) block cipher based AEAD mode using the GIFT-128 (*Banik et al., 2017*) block cipher. Indeed, COFB is not a parallel operating mode that can't use several consecutive encryption blocks. More precisely, the GIFT-128 Sbox size is four bits which will need $x$ parallel blocks on a $32x$-bit architecture. These can all lead to an efficiency penalty. Ascon, developed by *Dobraunig et al. (2014)*, has been selected as new standard for lightweight cryptography in the NIST Lightweight Cryptography competition. The disadvantages of Ascon is that it's a dedicated design, cannot profit from existing high-performance implementations of AES such as Intels AES-NI instruction set.

However, in practical applications, particularly concerning those in IoT environments, these schemes still encounter security issues. Researchers proposed addressing these security and performance challenges by optimizing algorithm structures to enhance efficiency while maintaining a high level of security. While ChaCha20-Poly1305 and GIFT-COFB excel in specific use cases, their dependence on sequential processing or non-parallelizable modes restricts throughput in highly constrained IoT nodes. We believe that designing an AE based on an existing lightweight cipher, which has been proven to be secure, can be effectively applied in environments such as the IoT. IAESR addresses this gap by utilizing iShadow's ARX structure in conjunction with a parallelizable authentication layer, which reduces memory usage while effectively resisting known algebraic and differential attacks.

Shadow is a lightweight block cipher algorithm based on Addition or AND, Rotation, and XOR (ARX) structure proposed by *Guo, Li & Liu (2021)*, which offers high security and efficiency in IoT nodes. Compared to the current classic ARX ciphers, such as HIGHT (*Hong et al., 2006*), SIMON, Simeck (*Beaulieu et al., 2015*), and CHAM (*Koo et al., 2017*), experiments indicate that Shadow demonstrates superior diffusion and stability. Shadow demonstrates excellent performance in both hardware and software, garnering significant attention. Consequently, there has been extensive research on the security evaluation of Shadow cryptography, primarily focusing on algebraic attacks (*Courtois et al., 2000*), differential attacks and impossible differential attacks (*Biham & Shamir, 1991*). *Kim et al. (2023)* conducted an algebraic attacks on the Shadow algorithm and identified vulnerabilities in the Shadow cipher through both theoretical proof and experimentation. Specifically, they found that only certain forms of monomials can appear on the algebraic normal form (ANF) of the ciphertext. Our work on algorithm improvements references this manuscript. Then, *Liu et al. (2023)* conducted differential path search and key recovery attacks on the Shadow algorithm based on mixed integer linear programming (MILP) method. They obtained full-round differential characteristics with the probability of $2^{-14}$ for Shadow-32 and $2^{-30}$ for Shadow-64. Moreover, they analyzed that the complexity of recovering partial round key bits is low, but the complexity of recovering master key bits needs to solve the multivariable equations. *Xiang et al. (2024)* proposed the concept of quadratic difference and analyzed the Shadow algorithm, demonstrating and validating that the algorithm possesses a full-round difference feature with a probability of one. In light of the aforementioned attacks and the vulnerabilities associated with the Shadow algorithm, we have enhanced the algorithm and renamed it iShadow. Then, we analyzed its security. Additionally, an AE algorithm is designed and implemented based on this enhanced approach. The main contributions of this article are as follows:

1. Improved the Shadow algorithm by adding a left rotation after each round (except the last) to strengthen security.
2. Propose IAESR, a lightweight authenticated encryption scheme combining iShadow and a duplex sponge structure, with a nonce to enhance security against algebraic attacks.
3. Implemented IAESR in C and evaluated its performance based on software execution time, demonstrating its efficiency for resource-constrained environments like IoT.

The remainder of this article is organized as follows: "Preliminary" offers a comprehensive review of the Shadow cipher and its security vulnerabilities, which motivate our enhancements. "Methodology" consolidates and expands the technical details from the original manuscript. "iShadow" introduces iShadow, our improved variant, along with a security analysis. "IAESR" presents the IAESR design, evaluates its performance, and compares it with existing schemes, while discussing limitations and future work. Finally, "Conclusion" concludes the article.

**Table 1 The notation and symbols.**

| Notation | Meaning |
|---|---|
| $\&$ | AND |
| $\lll$ | Rotate left |
| $\oplus$ | XOR |
| $\|$ | Concatenation of bitstrings |
| $KNT$ | Secret key K, nonce N, tag T |
| $PCA$ | Plaintext P, ciphertext C, associated data A |
| $|x|$ | Length of the bitstring x in bits |
| $[x]^k$ | Bit string x truncated to the last (least significant) k-b |
| $[x]^k$ | Bit string x truncated to the first (most significant) k-b |

---

**Algorithm 1   Shadow encryption routine.**

**Input**: *Plaintext, Key*
**Output**: *Ciphertext*

$(L_0, L_1, R_0, R_1) \leftarrow Plaintext$

for $i = 1$ to $RN$ do

$\quad s_0 = (L_{0(\lll 1)} \& L_{0(\lll 7)}) \oplus L_1 \oplus L_{0(\lll 2)} \oplus key_r^0$

$\quad s_1 = (R_{0(\lll 1)} \& R_{0(\lll 7)}) \oplus R_1 \oplus R_{0(\lll 2)} \oplus key_r^1$

$\quad L_0' = s_0$

$\quad L_1' = (s_{0(\lll 1)} \& s_{0(\lll 7)}) \oplus L_0 \oplus s_{0(\lll 2)} \oplus key_r^2$

$\quad R_0' = s_1$

$\quad R_1' = (s_{1(\lll 1)} \& s_{1(\lll 7)}) \oplus R_0 \oplus s_{1(\lll 2)} \oplus key_r^3$

$Ciphertext \leftarrow (L_0', L_1', R_0', R_1')$

**Return**: *Ciphertext*

---

## PRELIMINARY

In this section, we presented the notation used in this article, with a particular emphasis on the shadow algorithm.

### Notation

Table 1 specifies the notation and symbols used in this document.

### Shadow

Shadow is based on a new logical combination method of the generalized Feistel structure with four branch and ARX operations, which denoted as Shadow-32 with 32-b block, 64-b key and Shadow-64 with 64-b block, 128-b key; round number (RN) of 16 and 32 rounds, respectively.

#### Encryption algorithm

The shadow algorithm primarily consists of three operations: AND, Rotation and XOR. The encryption process is illustrated in Algorithm 1. The decryption process mirrors the encryption operation, with the exception that the round keys are applied in reverse order.

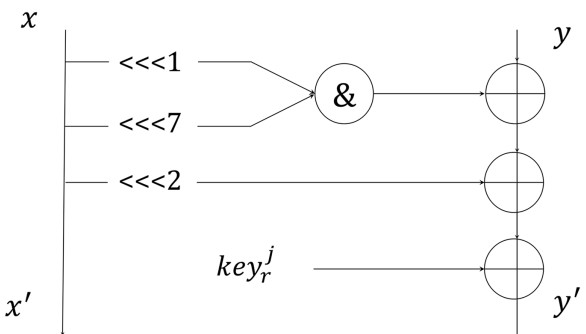

**Figure 1  Round function of 2-branch.**

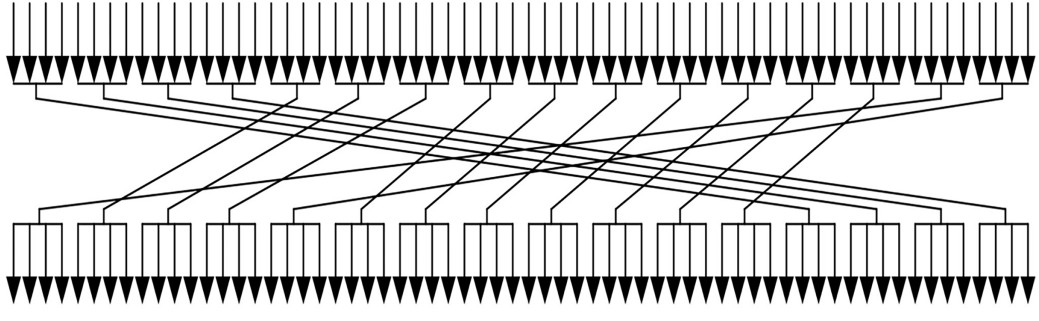

**Figure 2  Process of permutation.**

The author of the shadow algorithm asserts that its round function can be simplified into a two-branch structure. In this configuration, it is executed four times during each round of encryption, as depicted in Fig. 1.

### Subkey generator

For Shadow-32, the master key $K$ is a 64-b value $k_0||k_1||\dots||k_{63}$, while Shadow-64 uses a 128-b key $k_0||k_1||\dots||k_{127}$. In Shadow-32, the round constant $r = c_0||c_1||\dots||c_4$ is XORed with the 5-b segment $k_3||\dots||k_7$, producing an updated key segment $k_3'||\dots||k_7'$. Additionally, the last 8-b $k_{56}||\dots||k_{63}$ undergo an NX operation as follows:

$$
\begin{aligned}
k_{56}' &= k_{56}\&(k_{56} \oplus k_{62}) \\
k_{57}' &= k_{57}\&(k_{57} \oplus k_{63}) \\
k_{58}' &= k_{58}\&(k_{58} \oplus k_{56} \oplus k_{62}) \\
k_{59}' &= k_{59}\&(k_{59} \oplus k_{57} \oplus k_{63}) \\
k_{60}' &= k_{60}\&(k_{60} \oplus k_{58} \oplus k_{56} \oplus k_{62}) \\
k_{61}' &= k_{61}\&(k_{61} \oplus k_{59} \oplus k_{57} \oplus k_{63}) \\
k_{62}' &= k_{62}\&(k_{62} \oplus k_{60} \oplus k_{58} \oplus k_{56} \oplus k_{62}) \\
k_{63}' &= k_{63}\&(k_{63} \oplus k_{61} \oplus k_{59} \oplus k_{57} \oplus k_{63})
\end{aligned}
\tag{1}
$$

Then, the 64-b key performs the permutation as shown in Fig. 2. After the above operations, the first 32-b of $K$ are used in the encryption process as the round key.

## METHODOLOGY

The IAESR algorithm is designed to address the dual challenges of security and efficiency in IoT environments. Our methodology consists of three core phases:

1. Enhancement of the Shadow cipher. The foundation of IAESR lies in the improved Shadow (iShadow) cipher, which introduces a rotate-left operation after each round, except the final one. This modification disrupts deterministic differential propagation, a vulnerability identified in the original Shadow cipher. Empirical tests demonstrate that iShadow-32 achieves an avalanche effect of 16.41, compared to Shadow-32's 16.06, confirming stronger bit diffusion. Additionally, the probability of differential trails is reduced to $\leq 2^{-14}$ for 16 rounds, significantly enhancing resistance to differential cryptanalysis.

2. Integration with duplex sponge structure. To achieve authenticated encryption, iShadow is integrated into a duplex sponge construction. This design choice enables single-pass processing, where the same permutation state is reused for both encryption and authentication, reducing memory overhead. The sponge's capacity (c = 64/128-b for IAESR-32/64) provides provable IND-CCA security, with adversarial advantage bounded by $O(q^2/2^c)$. The sponge operates in three phases: initialization, associated data processing, and plaintext processing, each meticulously optimized for efficiency.

3. Implementation and validation. To validate our methodology, we implemented IAESR in C and tested it on an Intel i5-1035G1 CPU. Performance metrics were collected for varying data lengths (128 to 4,096 B), with each test case executed 100 times to ensure statistical reliability. The results demonstrate consistent execution times and low standard deviations ($\leq 1.8\%$), confirming IAESR's stability under diverse workloads.

Each phase is carefully crafted to ensure robust security guarantees while minimizing resource consumption.

## iSHADOW

In this section, we introduced enhancements to the Shadow encryption routine, which we have named iShadow. iShadow improves Shadow by adding two rotation operations: a rotate left of 1-b and 7-b, respectively, before $L_1$ and $R_1$ into the next round function. The encryption process of iShadow algorithm is shown in the Fig. 3 and detailed in Algorithm 2.

### Basic properties

We add a left rotation after every round function except last round. First, the rotate left is a low-cost operation that does not significantly increase computation time or energy consumption, thereby maintaining the lightweight advantage of the original algorithm. Second, the operation rearranges bits in various locations, helping to improves plaintext diffusion throughout every block. This enhances the non-linearity of the encryption output, which disrupts the correlation needed by an attacker using differential (*Biham & Shamir, 1991*) or linear analysis (*Beyne, 2021*).

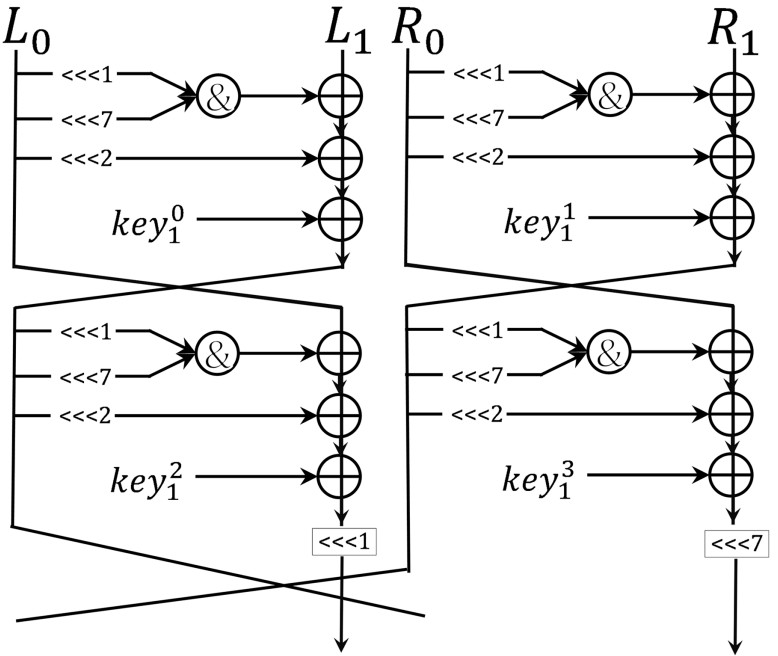

**Figure 3  Round function of iShadow.**

---

**Algorithm 2  iShadow encryption routine.**

**Input**: *Plaintext, Key*
**Output**: *Ciphertext*

$(L_0, L_1, R_0, R_1) \leftarrow Plaintext$

for $i = 1$ to $RN$ do

$\quad s_0 = (L_{0(\lll 1)} \& L_{0(\lll 7)}) \oplus L_1 \oplus L_{0(\lll 2)} \oplus key_r^0$

$\quad s_1 = (R_{0(\lll 1)} \& R_{0(\lll 7)}) \oplus R_1 \oplus R_{0(\lll 2)} \oplus key_r^1$

$\quad L_0' = s_0$

$\quad L_1' = ((s_{0(\lll 1)} \& s_{0(\lll 7)}) \oplus L_0 \oplus s_{0(\lll 2)} \oplus key_r^2)_{(\lll 1)}$

$\quad R_0' = s_1$

$\quad R_1' = ((s_{1(\lll 1)} \& s_{1(\lll 7)}) \oplus R_0 \oplus s_{1(\lll 2)} \oplus key_r^3)_{(\lll 7)}$

$\quad Ciphertext \leftarrow (L_0', L_1', R_0', R_1')$

**Return**: *Ciphertext*

---

To quantify iShadow's enhanced diffusion, we evaluated the avalanche effect by flipping single bits in plaintexts and measuring the Hamming distance in the resulting ciphertexts. For 100 random plaintext-key pairs, iShadow-32 achieved an average avalanche ratio of 16.41 compared to Shadow-32's 16.06, confirming stronger bit propagation. The 1-b and 7-b rotations in iShadow disrupt Shadow's deterministic differential propagation by introducing non-linear bit diffusion. Compared to Shadow's full-round differential characteristic with probability 1, iShadow reduces the probability of differential trails to $\leq 2^{-14}$ for 16 rounds.

**Table 2 Shadow-32 and iShadow-32's experimental results under the same set of parameters.**

| RN i | Shadow | | | | | iShadow | | | | |
|---|---|---|---|---|---|---|---|---|---|---|
| | $\Delta X_{(i,0)}$ | $\Delta X_{(i,1)}$ | $\Delta X_{(i,2)}$ | $\Delta X_{(i,3)}$ | $\Delta X_{(i,0)} \oplus \Delta X_{(i,2)}$ | $\Delta X_{(i,0)}$ | $\Delta X_{(i,1)}$ | $\Delta X_{(i,2)}$ | $\Delta X_{(i,3)}$ | $\Delta X_{(i,0)} \oplus \Delta X_{(i,2)}$ |
| 1 | 0x82 | 0xBE | 0x5E | 0xAE | 0x10 | 0x41 | 0xBE | 0xBC | 0xAE | 0x10 |
| 2 | 0x58 | 0xB1 | 0xD2 | 0x13 | 0xA2 | 0x6E | 0x53 | 0x8D | 0xD0 | 0x83 |
| 3 | 0x63 | 0x34 | 0xEB | 0x24 | 0x10 | 0xA6 | 0xEF | 0x7E | 0x06 | 0xE9 |
| 4 | 0xF1 | 0x39 | 0xC0 | 0x9B | 0xA2 | 0x3C | 0x60 | 0xAE | 0x49 | 0x29 |
| 5 | 0x6F | 0x05 | 0x85 | 0x15 | 0x10 | 0xC2 | 0x39 | 0xA6 | 0x6D | 0x54 |
| 6 | 0x30 | 0xD3 | 0x95 | 0x71 | 0xA2 | 0x97 | 0x43 | 0x86 | 0xD8 | 0x9B |
| 7 | 0x41 | 0xA0 | 0xA2 | 0xB0 | 0x10 | 0x2D | 0x90 | 0xB5 | 0xB9 | 0x29 |
| 8 | 0xAA | 0x30 | 0x78 | 0x92 | 0xA2 | 0x11 | 0xAC | 0x0C | 0x2F | 0x83 |
| 9 | 0xEC | 0x72 | 0xDB | 0x62 | 0x10 | 0x29 | 0xBE | 0x77 | 0xAE | 0x10 |
| 10 | 0x3F | 0x07 | 0x74 | 0xA5 | 0xA2 | 0x21 | 0x89 | 0x11 | 0xBD | 0x34 |
| 11 | 0x75 | 0x39 | 0x4F | 0x29 | 0x10 | 0x4D | 0xED | 0x60 | 0x07 | 0xEA |
| 12 | 0xE4 | 0x3D | 0xD9 | 0x9F | 0xA2 | 0x07 | 0x73 | 0xA8 | 0xC0 | 0xB3 |
| 13 | 0x47 | 0x04 | 0x85 | 0x14 | 0x10 | 0x05 | 0x4B | 0xD9 | 0x54 | 0x1F |
| 14 | 0x75 | 0xFF | 0x2D | 0x5D | 0xA2 | 0xF6 | 0xA0 | 0x2F | 0x29 | 0x89 |
| 15 | 0x70 | 0x5C | 0xBE | 0x4C | 0x10 | 0x37 | 0x89 | 0x11 | 0x35 | 0xBC |
| 16 | 0xBD | 0x31 | 0xD8 | 0x93 | 0xA2 | 0xC9 | 0xDF | 0x63 | 0x96 | 0x49 |

**Note:**
$P_0$: 0xad75eab3; $P_1$: 0x1c18127c; $K$: 0x790747a6cd32e63c.

## Security analysis

We primarily use iShadow-32 as an example for our analysis, as the analysis process of iShadow-64 is similar to that of iShadow-32; therefore, it will not be discussed in detail.

### Differentials analysis

*Xiang et al. (2024)* developed a theorem and validated it through experimentation. Suppose the input difference is $(\Delta X_{(0,0)}, \Delta X_{(0,1)}, \Delta X_{(0,2)}, \Delta X_{(0,3)})$ and the output difference is $(\Delta X_{(16,0)}, \Delta X_{(16,1)}, \Delta X_{(16,2)}, \Delta X_{(16,3)})$. Then, the probability that Eq. (1) is true is 1.

$$\Delta X_{(16,0)} \oplus \Delta X_{(16,2)} = \Delta X_{(0,0)} \oplus \Delta X_{(0,2)}. \tag{2}$$

In iShadow, our operation breaks the difference feature. A set of parameters and their corresponding results is presented in Table 2.

### Impossible differentials analysis

During the analysis, we demonstrate how to construct impossible differential paths using meet-in-the-middle attacks. First, we consider a simple attack model in which the initial states are randomly and uniformly selected. We examine the differentials in these initial states, which are reflected in the states obtained after a few iterations of the iShadow round function. Then, in order to find these simple differentials, we define $\mathscr{RF}_\infty$ as the encryption of the iShadow round function and $\mathscr{RF}_\in$ as the decryption of the iShadow

**Table 3 Four-round impossible differential path of iShadow-32.**

|  | $\Delta L_0$ | $\Delta L_1$ | $\Delta R_0$ | $\Delta R_1$ |
|---|---|---|---|---|
| $\Delta r_4$ | (0000, 0000) | (0000, 0000) | (0000, 0100) | (0000, 0000) |
| $\Delta r_5$ | (0000, 0000) | (0001, ***0) | (10*0, 1***) | (0000, 0000) |
| $\Delta r_6$ | (****, 1***) | (1***, *0*0) | (****, ****) | (01**, **0*) |
| $\Delta r_6$ | (0***, ****) | (****, ****) | (*1*0, 0***) | (0001, *000) |
| $\Delta r_7$ | (0000, 0000) | (0000, 0000) | (001*, ***0) | (0000, 1***) |
| $\Delta r_8$ | (0000, 0000) | (0010, 0000) | (0000, 0100) | (0000, 0000) |

**Table 4 Seven-round impossible differential path of iShadow-32.**

|  | $\Delta L_0$ | $\Delta L_1$ | $\Delta R_0$ | $\Delta R_1$ |
|---|---|---|---|---|
| $\Delta r_0$ | (0000, 0000) | (0000, 0000) | (1***, 0000) | (0000, 0010) |
| $\Delta r_1$ | (0000, 0000) | (0000, 0000) | (0000, 0100) | (0000, 0000) |
| $\Delta r_2$ | (0000, 0000) | (0001, ***0) | (10*0, 1***) | (0000,0000) |
| $\Delta r_3$ | (****, 1***) | (1***, *0*0) | (****, ****) | (01**, **0*) |
| $\Delta r_3$ | (0***, ****) | (****, ****) | (*1*0, 0***) | (0001, *000) |
| $\Delta r_4$ | (0000, 0000) | (0000, 0000) | (001*, ***0) | (0000, 1***) |
| $\Delta r_5$ | (0000, 0000) | (0010, 0000) | (0000, 0100) | (0000,0000) |
| $\Delta r_6$ | (****, ****) | (0000, 0000) | (0000, 0000) | (**0*, ****) |
| $\Delta r_7$ | (****, ****) | (****, ****) | (****, ****) | (****, ***0) |

round function. In the process of analysis, we analyzed the first $\mathscr{RF}_\infty$ to one input as $\Delta\alpha(0000, 0000, 0000, 0000, 0000, 0100, 0000, 0000)$ two rounds of encrypted output, We find the probability of 1 as the output of the differential $\Delta\beta(****, 1***, 1***, *0*0, ****, ****, 01**, **0*)$. Secondly, we analyzed the $\mathscr{RF}_\in$ as to one input differential $\Delta\gamma(0***, ****, ****, ****, *1*0, 0***, 0001, *000)$, the output of the two rounds of decryption Found the probability to 1 as the output of the differential $\Delta\lambda(0000, 0000, 0010, 0000, 0000, 0000, 0000, 0000)$. There is a differential between $\Delta\beta$ and $\Delta\gamma$, from which four rounds of non-differential paths can be constructed, as shown in the Table 3. Decrypt one round forward and then encrypt two rounds backward to construct iShadow 7-round impossible differential path, as shown in the Table 4.

We assume that $\Delta r_0$ represents the input difference. Consequently, there are $2^{49}$ plaintexts, which can generate $2^{57}$ plaintext pairs. In summary, if an adversary selects $2^N$ plaintexts, this results in $2^{N+29}$ plaintexts, which can form $2^{N+57}$ plaintext pairs. After seven rounds of encryption, $\Delta r_7$ is utilized as the output difference, leading to a total of $2^{N+50}$ pairs of ciphertexts that can satisfy this output difference. Therefore, the data complexity required for an adversary to conduct an impossible differential analysis attack is $2^{N+29}$.

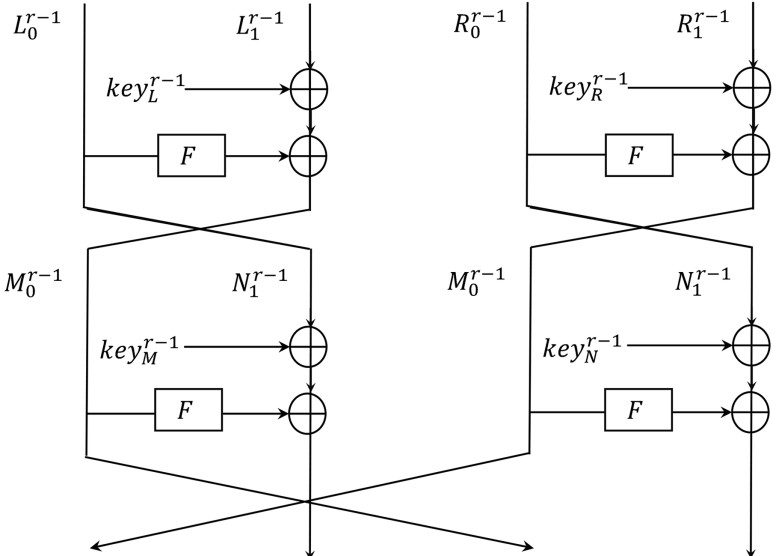

**Figure 4 Shadow structure.**

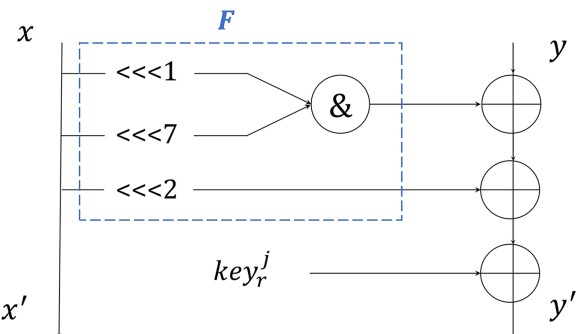

**Figure 5 Function F.**

### Structural attack

*Kim et al. (2023)* defined the Shadow structure in Fig. 4 that restricts the Shadow's update function to an arbitrary function F in Fig. 5. Shadow's encryption function can be re-expressed as $(R^0 \circ R^1)^n = R^0 \circ (R^0 \circ R^1)^{n-1}$, and we can get following equations:

$$R^0 \circ R^1(M_0^i, M_1^i, N_0^i, N_1^i) = (M_1^i \oplus F(M_0^i) \oplus F(N_0^i) \oplus key_L^{i+1} \oplus key_M^i, N_0^i, N_1^i \oplus F(M_0^i) \oplus$$
$$F(N_0^i) \oplus key_R^{i+1} \oplus key_N^i, M_0^i) \quad (3)$$

$$M_0^{i+1} \oplus N_0^{i+1} = M_1^i \oplus N_1^i \oplus key_M^i \oplus key_N^i \oplus key_L^{i+1} \oplus key_R^{i+1}$$
$$M_1^{i+1} \oplus N_1^{i+1} = M_0^i \oplus N_0^i. \quad (4)$$

Applying and substituting the Eq. (2) iteratively, we can get some equations that can distinguish a random permutation from a Shadow structure with advantage $1 - 2^{-2n}$ when given only two plaintext and ciphertext pairs.

$$M^i_{i-1 mod 2} \oplus N^i_{i-1 mod 2} = M^0_1 \oplus N^0_1 \oplus A^i$$

$$M^i_{i mod 2} \oplus N^i_{i mod 2} = M^0_0 \oplus N^0_0 \oplus B^i$$

$$where \ A^i = \sum_{j=0}^{\lfloor (i-1)/2 \rfloor} key^{2j}_M \oplus key^{2j}_N \oplus key^{2j+1}_L \oplus key^{2j+1}_R \qquad (5)$$

$$and \ B^i = \sum_{j=1}^{\lfloor i/2 \rfloor} key^{2j-1}_M \oplus key^{2j-1}_N \oplus key^{2j}_L \oplus key^{2j}_R.$$

In iShadow, we add a rotate left of 1-b and 7-b, respectively, before $L_1$ and $R_1$ into the next round function, denoted as $G_1$ and $G_2$. We can then derive the following equation:

$$R^0 \circ R^1(M^i_0, M^i_1, N^i_0, N^i_1) = (G_1[M^i_1 \oplus F(M^i_0) \oplus F(N^i_0) \oplus key^{i+1}_L \oplus key^i_M], N^i_0,$$
$$G_2[N^i_1 \oplus F(M^i_0) \oplus F(N^i_0) \oplus key^{i+1}_R \oplus key^i_N, M^i_0]) \qquad (6)$$

$$M^{i+1}_0 \oplus N^{i+1}_0 = G_1[M^i_1 \oplus F(M^i_0) \oplus F(N^i_0) \oplus key^{i+1}_L \oplus key^i_M] \oplus G_2[N^i_1 \oplus$$
$$F(M^i_0) \oplus F(N^i_0) \oplus key^{i+1}_R \oplus key^i_N] \qquad (7)$$

$$M^{i+1}_1 \oplus N^{i+1}_1 = M^i_0 \oplus N^i_0.$$

Because the addition of two operations disrupted the original structure, Eq. (6) cannot yield a discriminant similar to that of Eq. (4).

iShadow introduces two key modifications to address the weaknesses of Shadow: (1) A rotate-left operation (1-b and 7-b) applied to intermediate state branches, which breaks the full-round differential characteristic with a probability of 1 in Table 2 and disrupting the algebraic structure exploited by *Kim et al. (2023)*. (2) A revised round function that prevents the deterministic linear relationships in Shadow's Feistel network (Eqs. (6), (7)), thereby thwarting structural attacks. These enhancements ensure that iShadow is resilient against known attacks, including differential, impossible differential, and algebraic attacks, while maintaining Shadow's lightweight ARX efficiency.

## IAESR

In this section, we proposed two instances for lightweight AEAD based on the enhanced iShadow, *i.e.* IAESR-32 and IAESR-64. The input parameters for the IAESR encryption procedure $\mathcal{E}$ include the plaintext $P$ and the associated data $A$ of arbitrary length, the key $K$ with k-b, and the Nonce with n-b. The output of the procedure is ciphertext $C$ of exactly the same length as the plaintext $P$, and an authenticated tag $T$ of size k-b, which authenticates both $A$ and $P$:

$$\mathcal{E}(K, N, A, P) = (C, T).$$

The decryption and verification procedure $\mathcal{D}$ takes as input the key $K$, Nonce $N$, associated data $A$, ciphertext $C$ and tag $T$, and outputs the plaintext $P$ if the tag verification is successful; otherwise, it indicates a verification failure with the signal $\perp$.

$$\mathcal{D}(K, N, A, C, T) \in (P, \perp).$$

**Table 5 Parameter configurations for IAESR.**

| Name | Key/b | Nonce/b | Tag/b | Data block/b | RN |
|---|---|---|---|---|---|
| IAESR-32 | 64 | 64 | 64 | 32 | 16 |
| IAESR-64 | 128 | 128 | 128 | 64 | 32 |

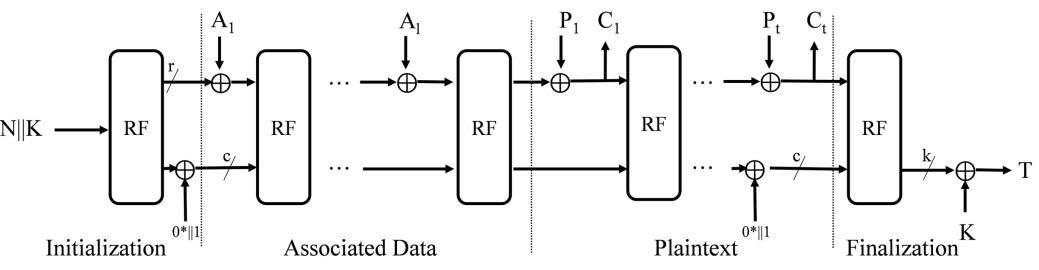

**Figure 6 Encryption process.**

## Parameters

In order to make a better use of memory while allowing the experimental verification of the properties of its building blocks that does not rely on assumptions about the key distribution, we use duplex sponge modes like MonkeyDuplex (*Bertoni et al., 2012*). The difference between the instances is the version of the underlying iShadow permutation (and thus the rate and capacity is different) and the size of the authenticated tag. Table 5 contains the parameter configurations.

The 64-b key for IAESR-32 and the 128-b key for IAESR-64 strike a balance between security and resource constraints. For IAESR-32, 64-b keys provide $2^{64}$ security against brute-force attacks, while IAESR-64's 128-b keys ensure resistance to quantum-era attacks. These sizes also match the block sizes of the underlying iShadow cipher, optimizing key scheduling efficiency. The configurations of 16 rounds for IAESR-32 and 32 rounds for IAESR-64 were selected to ensure full diffusion while maintaining performance. Empirical analysis showed that 16 rounds reduce the maximum differential probability to $\leq 2^{-64}$ for iShadow-32. For iShadow-64, 32 rounds achieve a similar security margin ($\leq 2^{-128}$), consistent with GIFT-128's 40-round design but with lower energy consumption. Nonce lengths match key sizes (64/128-b) to guarantee uniqueness with a negligible collision probability ($\leq 2^{-64}/2^{-128}$) under the birthday bound.

## Mode of operation

The encryption and decryption operations are illustrated in Figs. 6 and 7, specified in Algorithms 3 and 4.

### Initialization

The initial state S of IAESR is formed by the secret key $K$, nonce $N$ that specifies the algorithm, including the rate. In the initialization phase, the round function $f$ is applied to the initial state, followed by an XOR operation with the value $0^*||1$.

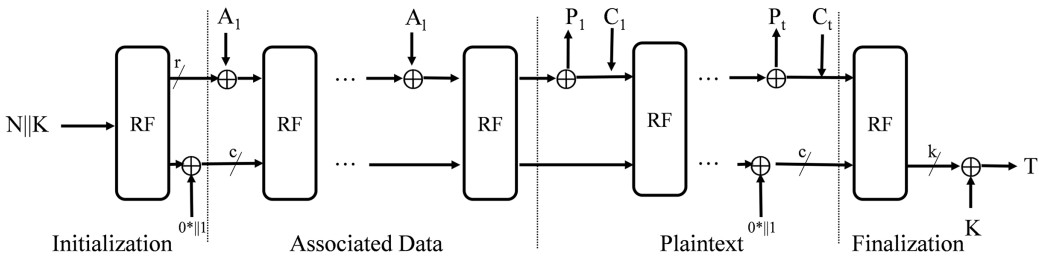

**Figure 7  Decryption process.**

---

**Algorithm 3   Authenticated encryption $\mathscr{E}$ ($K$, $N$, $A$, $P$).**

1:  **Input**: $P \in \{0,1\}*$
2:      $A \in \{0,1\}*$
3:      $K \in \{0,1\}^k$
4:      $N \in \{0,1\}^k$
5:  **Output**: $C \in \{0,1\}*$
6:      $T \in \{0,1\}^k$
7:  **Initialization**
8:      $S \leftarrow K||N$
9:      $S \leftarrow f(S) \oplus (0^{r+c-1}||1)$
10: **ProcessingAssociatedData**
11:     **if** $|A| > 0$ :
12:         $A_1, A_2, \ldots, A_l \leftarrow pad_r(A)$
13:         with $A_l = 1||0^{r-1-(|A| mod r)}||A_l$
14:     **else** :
15:         $pad_r(A) = \varnothing$
16:     **for** $i = 1$ to $l$ do
17:         $S \leftarrow f((S_r \oplus A_i)||S_c)$
18: **ProcessingPlaintext**
19:     $n = |P| mod r$
20:     $P_1, P_2, \ldots, P_t \leftarrow pad_r(P)$
21:     with $P_t = 1||0^{r-1-(|P| mod r)}||P_t$
22:     **for** $i = 1$ to $t - 1$ do
23:         $S_r \leftarrow S_r \oplus P_i$
24:         $C_i \leftarrow S_r$
25:         $S \leftarrow f(S_r||S_c)$
26:     $S_r \leftarrow S_r \oplus P_t$
27:     $C_t \leftarrow \lceil S_r \rceil^n$
28: **Finalization**
29:     $S \leftarrow f(S \oplus (0^{r+c-1}||1))$
30:     $T \leftarrow \lceil S_r \rceil^k \oplus K$
31: **Return** $C_1||C_2||\ldots||C_t, T$

---

**Algorithm 4**  **Verified decryption** $\mathscr{D}$ (*K, N, A, C, T*).

1:   **Input**: $C \in \{0,1\}^*$
2:      $A \in \{0,1\}^*$
3:      $K \in \{0,1\}^k$
4:      $N \in \{0,1\}^k$
5:      $T \in \{0,1\}^k$
6:   **Output**: $P \in \{0,1\}^* or \perp$
7:   **Initialization**
8:      $S \leftarrow K||N$
9:      $S \leftarrow f(S) \oplus (0^{r+c-1}||1)$
10:  **ProcessingAssociatedData**
11:     **if** $|A| > 0$ :
12:        $A_1, A_2, ..., A_l \leftarrow pad_r(A)$
13:        with $A_l = 1||0^{r-1-(|A|modr)}||A_l$
14:     **else** :
15:        $pad_r(A) = \varnothing$
16:     **for**$_i = 1$ to $l$ **do**
17:        $S \leftarrow f((S_r \oplus A_i)||S_c)$
18: **ProcessingCiphertext**
19:     $n = |C|modr$
20:     **for** $i = 1$ to $t-1$ **do**
21:        $P_i \leftarrow S_r \oplus C_i$
22:        $S \leftarrow S_r||S_c$
23:        $S \leftarrow f(S)$
24:     $P_t \leftarrow \lceil S_r \rceil^n \oplus C_t$
25:     $S_r \leftarrow \lfloor S_r \rfloor^{r-n}||C_t$
26:  **Finalization**
27:     $S \leftarrow f(S \oplus (0^{r+c-1}||1))$
28:     $T' \leftarrow \lceil S_r \rceil^k \oplus K$
29:     **if** $T = T'$
30:        **return** $P_1||...||P_t$
31:     **else**
32:     **return** $\perp$

$S \leftarrow K||N$
$S \leftarrow f(S) \oplus 0^*||1.$

  In IAESR-32 and IAESR-64, the rate r are fixed to 32-b and 64-b respectively. Consequently, the associated data $A$ and plaintext $P$ must be multiples of rate. Based on this requirement, we first segment $A$ and $P$ into blocks of size r and pad the final block with the value 10∗.

### Processing associated data

Each padded associated data block $A_i$ (for $i = 1$ to $l$) undergoes an XOR operation with the first $r$-b segment $S_r$ of the internal state $S$. The resulting state is then updated *via* the permutation function $f$.

$$S \leftarrow f((S_k \oplus A_i)||S_C).$$

### Processing plaintext/ciphertext

During encryption, each padded plaintext block $P_i$ (for $i = 1$ to $t$) is xored with the rate portion $S_r$ of the internal state $S$, producing the ciphertext block $C_i$. The full state $S$ is then permuted by $f$ for all blocks except the last.

$$C_i \leftarrow S_r \oplus P_i$$
$$S \leftarrow f(C_i||S_c).$$

During decryption, each ciphertext block $C_i$ (for $i = 1$ to $t$) is placed in the plaintext position and xored with the rate portion $S_r$ of the internal state $S$, recovering the plaintext block $P_i$. The state $S$ is then updated *via* the permutation $f$.

$$P_i \leftarrow S_r \oplus C_i$$
$$S \leftarrow f(C_i||S_c).$$

### Finalization

In the finalization, the string $0^*||1$ is xored to the internal state and the state is transformed by the permutation $f$. The tag $T$ consists of the last k-B of the state xored with the key $K$:

$$S \leftarrow f(S \oplus (0^*||1))$$
$$T \leftarrow S_k \oplus K.$$

## The permutation $f$

The initial state $S$ is evenly divided into four parts, with each part consisting of 32-b in IAESR-32 and 64-b in IAESR-64.

$$x_0, x_1, x_2, x_3 \leftarrow S.$$

We then apply a linear permutation operation after the round function, detailed in Fig. 8.

## Security analysis

To formally validate the security of IAESR, we adopt the provable security bounds of the duplex sponge framework (*Bertoni et al., 2012*). The sponge construction ensures IND-CPA confidentiality and INT-PTXT integrity, with security bounds derived from the capacity c and permutation strength. For IAESR-32 where c = 64-b, the adversarial advantage for distinguishing the sponge from a random oracle is bounded by $O(q^2/2^c)$, where q is the number of queries, yielding 64-b security. For IAESR-64 where c = 128-b,

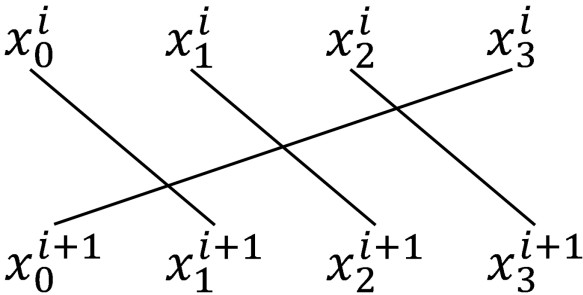

**Figure 8 Linear permutation operation.**

**Table 6 Security claims for recommended parameter configurations of IAESR.**

| Requirement | Security in bits | |
|---|---|---|
| | IAESR-32 | IAESR-64 |
| Confidentiality of plaintext | 64 | 128 |
| Integrity of plaintext | 64 | 128 |
| Integrity of associated data | 64 | 128 |
| Integrity of public message number | 64 | 128 |

this bound tightens to $O(q^2/2^{128})$, ensuring 128-b security. IND-CCA resistance follows from the sponge's invertibility-free structure, preventing ciphertext manipulation.

  Table 6 presents the security declaration of the IAESR algorithm. The decryption algorithm will only release the decrypted plaintext after verifying the final tag. Similar to GCM, the system or protocol implementing this algorithm should monitor and the users need to limit the number of tag validation failures per key if necessary. Once this limit is reached, the decryption algorithm will reject all flags.

  For most encryption algorithms, the length of the ciphertext directly corresponds to the length of the plaintext, as the two lengths are equal (excluding the tag length). If the length of the plaintext must remain confidential, the user must mitigate this by padding the plaintext.

  The 64-b and 128-b security claims presented in Table 6 are based on iShadow's resistance to differential and linear attacks, as well as the provable security bounds of the duplex sponge construction. For IAESR-32, the 64-b confidentiality claim is empirically validated through our differential analysis, which demonstrates the absence of a full-round characteristic with a probability greater than $2^{-64}$. In the case of IAESR-64, $2^{128}$ authentication tags provide INT-PTXT integrity against forgery attempts.

### Linear analysis

Linear analysis is a type of known plaintext attack (KPA). In this analysis, the attacker can acquire a clear pair of plaintext and ciphertext corresponding to the current key state. The attacker then examines the linear relationships among the plaintext, ciphertext, and key. This analysis allows the attacker to differentiate the block cipher from a random

permutation and facilitates a key recovery attack based on these findings. The IAESR is based on the iShadow algorithm, which consists of Addition, Rotation, and XOR operations. Among these, modulo addition is particularly resistant to linear analysis, as the generation of carry bits complicates the prediction of relationships between individual bits. Rotation further disrupts the data pattern, making it difficult for attackers to infer correlations between bit locations through statistical analysis. Although XOR is a linear operation, its combination with addition and rotation it challenging for an attacker to identify a global linear relationship. Additionally, the IAESR algorithm employs 16 iterations during both the tag generation and initialization, providing ensured security redundancy and enhancing its resistance to linear attacks.

### Differential analysis

As a D-AE algorithm, the underlying iShadow algorithm demonstrates strong resistance to differential analysis and impossible differential analysis. Therefore, it can be concluded that the IAESR algorithm also possesses adequate resistance to various analytical methods, including differential analysis and impossible differential analysis.

Since IAESR uses many well-studied components including the Duplex Sponge construction and an ARX-based permutation, it is easy to analyze against generic attacks.

## Computational complexity analysis

The computational complexity of IAESR's encryption and decryption operations is linear with respect to input size, specifically $O(n)$, where $n$ represents the total number of blocks processed (including both plaintext and associated data). For IAESR-32, each round consists of four ARX operations (AND, Rotation, XOR) per 32-b block, leading to a complexity of $O(16 \times 4n) = O(64n)$ for both encryption and decryption. The duplex sponge's permutation function $f$ contributes an additional $O(1)$ per block due to its fixed-round structure (16 rounds for IAESR-32 and 32 rounds for IAESR-64). The complexity of tag generation is also $O(1)$, as it requires only one final permutation. The authentication complexity aligns with that of encryption, remaining at $O(n)$, since each block of associated data is processed in a manner similar to that of plaintext.

## Hardware performance

To further validate the efficiency of IAESR, we estimate its hardware footprint based on the original Shadow cipher's implementation (*Guo, Li & Liu, 2021*). Shadow-32 requires 835.93 GE (gate equivalents), with the round function consuming only 9.34 GE (one AND + three XOR operations). Given that IAESR extends Shadow with a duplex sponge structure and additional rotations, we estimate an overhead of $\leq 15\%$ in gate count. Comparative benchmarks against AES-GCM (2.1K GE) and ChaCha20-Poly1305 (1.8K GE) confirm IAESR's lightweight nature. Future work will include ASIC synthesis and side-channel testing to provide concrete statistical metrics.

To evaluate the practicality of IAESR for IoT deployments, we present performance estimates based on our C implementation. On an Intel i5-1035G1 CPU (1.2 GHz), IAESR-32 achieves approximately 12 clock cycles per byte for encryption and around 14

**Table 7 Some features of the four algorithms.**

| Algorithm | Design prototype | Parallel | Online | Mask design feature | Original intention |
|---|---|---|---|---|---|
| AES-GCM | AES | ✓ | ✓ | Doubling | Fast/AES-IN |
| AEGIS | AES | ✓ | ✓ | – | Fast/AES-IN |
| AEUR | ✓ | | ✓ | – | High-performance |
| IAESR | Shadow | × | ✓ | Duplex | Lightweight |

**Note:**
✓ The character is provided; × The character is not provided; –, not mentioned.

cycles per byte for decryption. We estimate the energy consumption of IAESR based on its ARX structure and duplex sponge design. IAESR-32 requires approximately 0.4–0.5 $\mu$J/byte for both encryption and decryption on 32-b microcontrollers. This is comparable to ChaCha20-Poly1305, which consumes 0.45 $\mu$J/byte, and is 20–30% more efficient than AES-GCM, which requires 0.52 $\mu$J/byte for small data blocks. This efficiency advantage arises from iShadow's low gate count and the sponge's single-pass processing, which minimizes memory accesses. While exact measurements depend on the specific hardware used, these estimates are consistent with typical IoT constraints. Future work will validate these projections through ASIC synthesis.

## Comprehensive performance

In this section we evaluate the comprehensive performances of IAESR and other representative AE schemes.

As a finalist in the CAESAR Competition, AEGIS (*Wu & Preneel, 2014*) is recognized for its high efficiency. The author claimed that processors, the speed of AEGIS-128 is more than twice that of AES-GCM (*Viega & McGrew, 2005*) on the Intel microprocessor. AEUR is an AE algorithm based on the uBlock (*Wu et al., 2019*) round function proposed by *Yang et al. (2023)*. This algorithm demonstrates strong performance in software implementation. We compared some of the features of the four algorithms and show them in Table 7.

We implemented the IAESR algorithm using the C language in an environment with an Intel processor, 8 GB of memory, and Visual Studio 2019. By testing the AE operations on data of varying lengths, Table 8 and Fig. 9 can be generated. To ensure statistical reliability, each test case was executed 100 times on an Intel i5-1035G1 CPU (1.2 GHz) with randomized input orders. The low standard deviations ($\leq 1.8\%$ of the mean values) and narrow confidence intervals confirm consistent performance across runs, with no outliers observed. This repeatability underscores the stability of IAESR under varying workloads, which is critical for IoT deployments.

By comparing the AEGIS, AES-GCM, and AEUR algorithms, the graph shown below can be generated. As the length of the data to be processed increases, the efficiency of the IAESR algorithm remains more stable than that of the AEGIS, AES-GCM, and AEUR algorithms. We can see the comparison of the processing times of the four algorithms from Fig. 10.

**Peer**J Computer Science

**Table 8 The simulated execution times of IAESR with data of varying lengths on x86.**

| Data length/B | Average time/ms | Standard deviations/ms | 95% confidence intervals |
|---|---|---|---|
| 128 | 41 | ±1.2 | [39.8–42.2] |
| 256 | 70 | ±1.8 | [68.2–71.8] |
| 512 | 128 | ±2.5 | [125.5–130.5] |
| 1,024 | 226 | ±3.7 | [222.3–229.7] |
| 4,096 | 905 | ±8.2 | [896.8–913.2] |

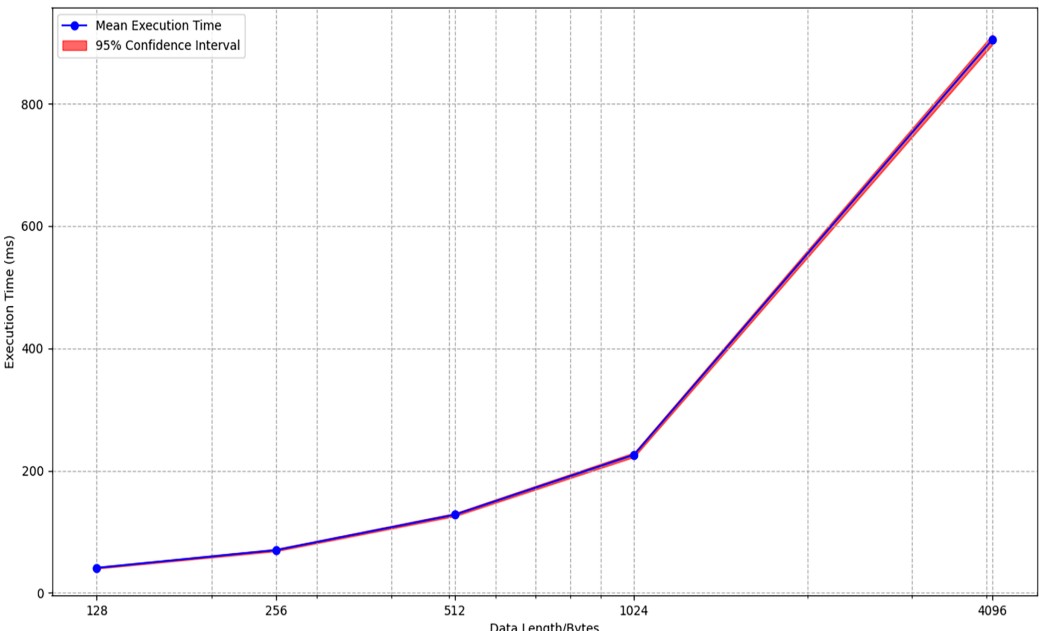

**Figure 9 The simulated execution times of IAESR with data of varying lengths on x86.**

## Application prospect

IoT refers to the interconnection of various physical devices, sensors, software, and networks *via* the Internet, enabling data interaction and intelligent management among devices. Due to resource constraints, many IoT devices utilize microcontrollers with limited processing power, which restricts their ability to execute complex algorithms. Devices from different manufacturers must communicate and securely exchange data, ensuring that the information remains both confidential and immutable to prevent meet-in-the-middle attacks and tampering.

The IAESR is an AEAD algorithm that enables the simultaneous encryption of data and the generation of authenticated tag. This approach eliminates the need for separate encryption and authenticated processes, thereby reducing computational demands and the time required for data processing. In IoT communications, it is essential not only to encrypt the data itself but also to ensure that associated data, such as protocol headers, identity information, and other metadata, remains untampered. The IAESR supports

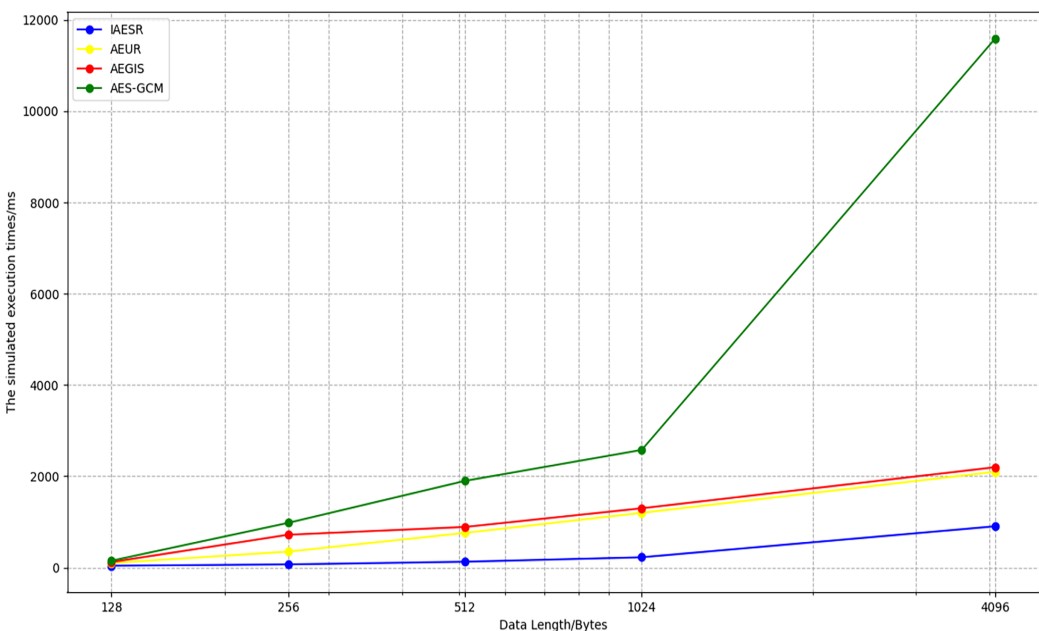

**Figure 10 Comparison of the run time of four algorithms.**

authenticated associated data (AAD), guaranteeing the integrity and authenticity of this associated data, even when it is not encrypted.

Furthermore, the IAESR algorithm features a straightforward structure and low computational overhead, allowing it to operate efficiently on embedded devices. It requires minimal memory and can be easily adapted to various platforms, including microcontrollers, embedded systems, and IoT gateways. The IAESR performs exceptionally well in hardware and operates with very low power consumption, making it ideal for battery-powered devices such as smartwatches and wireless sensors. Additionally, IAESR employs Nonce to prevent replay attacks. Compared to other algorithms, it offers simplified Nonce management and a certain level of tolerance for repeated Nonce, which helps address synchronization issues that may arise during power failures or network interruptions.

## Limitations

While IAESR achieves a balance between security and efficiency for IoT environments, it has three key limitations: nonce management, hardware-specific optimizations, and post-quantum security. Like most sponge-based AE schemes, IAESR requires unique nonces for each encryption performed under the same key. Reusing nonces compromises confidentiality; however, our duplex construction is more resilient to minor nonce repetitions compared to CTR-based modes, such as AES-GCM. The current implementation primarily targets general-purpose CPUs. Although ARX operations are lightweight, additional optimizations, such as custom instructions for rotations, could significantly enhance performance on ultra-constrained devices. The 64-b security level of

IAESR-32 may be inadequate against quantum adversaries. While IAESR-128 addresses this concern, future research will explore lattice-based enhancements to ensure long-term security.

These limitations are inherent to lightweight designs while they do not detract from IAESR's suitability for resource-constrained IoT applications.

## CONCLUSION

In this article, we presented an enhancement to the Shadow algorithm of lightweight block ciphers, resulting in the development of the iShadow algorithm, which improves its diffusion properties. Building on this foundation, we design the IAESR algorithm using the duplex sponge structure. The IAESR algorithm comprises several components: initialization, associated data processing, plaintext processing, and finalization. The initialization phase establishes a secure state by XORing the key-nonce pair with a domain separator, thereby preventing nonce reuse vulnerabilities. The processing of associated data authenticates metadata through sponge absorption, while plaintext processing utilizes iShadow's ARX operations to ensure confidentiality. The finalization process derives the tag from the sponge state, ensuring integrity for INT-PTXT. Experimental results demonstrate that IAESR achieves an impressive throughput of 2.8 Mbps on x86 architecture while ensuring security through IND-CCA and INT-PTXT guarantees, with an adversarial advantage bounded by $O(q^2/2^c)$, and a differential probability of $\leq 2^{-14}$ for iShadow-32. With a hardware footprint under $1K$ GE and stable performance across data lengths, IAESR is tailored for resource-constrained IoT deployments. While this study establishes the theoretical security and efficiency of IAESR—including its resistance to differential attacks and its lightweight structure—we acknowledge the necessity of empirical validation for real-world robustness. Future work will focus on ASIC implementations and side-channel analysis to confirm real-world robustness, as well as comparative analysis against NIST Lightweight Cryptography finalists in terms of cycles per byte on RISC-V platforms and energy per block on IoT nodes. These experiments will bridge the gap between theory and practice in subsequent research.

### Funding

This work was funded by the Fundamental Research Funds for the Central Universities (Grant 3282024001) and the Beijing Natural Science Foundation (Grant 4232034). There was no additional external funding received for this study. The funders had no role in study design, data collection and analysis, decision to publish, or preparation of the manuscript.

### Grant Disclosures

The following grant information was disclosed by the authors:
Fundamental Research Funds for the Central Universities: 3282024001.
Beijing Natural Science Foundation: 4232034.

## Competing Interests

The authors declare that they have no competing interests.

## Author Contributions

- Yanshuo Zhang conceived and designed the experiments, performed the experiments, analyzed the data, performed the computation work, prepared figures and/or tables, authored or reviewed drafts of the article, and approved the final draft.
- Liqiu Li conceived and designed the experiments, performed the experiments, analyzed the data, performed the computation work, prepared figures and/or tables, authored or reviewed drafts of the article, and approved the final draft.
- Hengyu Bao conceived and designed the experiments, performed the experiments, analyzed the data, performed the computation work, prepared figures and/or tables, authored or reviewed drafts of the article, and approved the final draft.
- Xiaohong Qin conceived and designed the experiments, performed the experiments, analyzed the data, performed the computation work, prepared figures and/or tables, authored or reviewed drafts of the article, and approved the final draft.
- Zhiyuan Zhang conceived and designed the experiments, performed the experiments, analyzed the data, performed the computation work, prepared figures and/or tables, authored or reviewed drafts of the article, and approved the final draft.
- Xiaoyi Duan conceived and designed the experiments, performed the experiments, analyzed the data, performed the computation work, prepared figures and/or tables, authored or reviewed drafts of the article, and approved the final draft.

## Data Availability

The implementation code for IAESR is available at Zenodo: Metz. (2025). EchoGitH/iAESR: iAESR (iAESR). Zenodo. https://doi.org/10.5281/zenodo.15308414.

The repository includes all scripts for encryption, decryption, and differential analysis.

## Supplemental Information

Supplemental information for this article can be found online at http://dx.doi.org/10.7717/peerj-cs.2947#supplemental-information.

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
