# Peer review of "IAESR: IoT-oriented authenticated encryption based on iShadow round function"

_PeerJ Computer Science, doi:10.7717/peerj-cs.2947_

## Round 0.1 · original submission · Major Revisions

Thank you for submitting your manuscript to PeerJ Computer Science. Based on the reviews, your paper addresses a relevant and promising area—lightweight authenticated encryption (AE) for IoT devices—but it requires significant revision before being considered for publication.

The reviewers have highlighted several critical areas that must be addressed comprehensively.

Empirical Validation:

The current version lacks sufficient experimental data to substantiate the claims of security, performance, and efficiency. Empirical validation is needed through practical tests and detailed experimental results, including statistical analysis and variance measures.

Include hardware implementation benchmarks (FPGA or ASIC), power consumption analysis, and detailed computational complexity evaluations to demonstrate real-world applicability, especially given the IoT context.

Security Analysis:

To solidify your security claims, provide formal security proofs under standard cryptographic models such as IND-CPA, IND-CCA, or INT-PTXT. Justify the selected security parameters (e.g., round numbers, key sizes) with rigorous analysis.

Comparative Analysis and Justification of Novelty:

Clearly articulate how your proposed algorithm advances state-of-the-art technology compared to lightweight AE solutions. Provide detailed comparative analysis and experimental benchmarks against relevant algorithms, highlighting improvements in diffusion, resistance to cryptanalysis, and efficiency.

Clarity, Structure, and Presentation:

Improve readability and clarity of technical explanations, breaking down overly complex sentences and ensuring terminology consistency.

Enhance the presentation of figures and tables, adding error bars and detailed captions for better interpretability.

Expand the literature review to demonstrate the knowledge gap your research addresses clearly.

Data Availability:

Provide access to raw data and experimental code per our journal's data availability policy. This is critical for independent verification of your results.

Addressing these major concerns is necessary for reconsideration. Your revisions should comprehensively respond to all reviewers’ comments, clearly detailing changes made and providing thorough justifications for methodological and analytical choices.

We look forward to receiving your revised manuscript.

**Language Note:** The review process has identified that the English language must be improved. PeerJ can provide language editing services - please contact us at [email protected] for pricing (be sure to provide your manuscript number and title). Alternatively, you should make your own arrangements to improve the language quality and provide details in your response letter. – PeerJ Staff

Reviewer 1 ·

Basic reporting

1. Language and Clarity: The paper is written in professional English, but there are areas where the language could be improved for better clarity, particularly in the technical descriptions of the algorithms and security analysis. Some sentences are overly complex and could be simplified for better comprehension by an international audience.
2. Introduction and Background: The introduction provides a good context for the research, highlighting the need for lightweight authenticated encryption (AE) algorithms in IoT environments. However, the literature review could be more comprehensive. While some relevant works are mentioned, the paper does not sufficiently discuss how this research fills a specific knowledge gap compared to existing solutions.
3. Structure: The structure of the paper generally conforms to PeerJ standards, but the flow of information could be improved. The transition between sections, particularly from the introduction to the technical details, is abrupt. Additionally, the paper lacks a clear discussion of the limitations of the proposed solution.
4. Figures and Data: The figures are relevant and well-labeled, but some of the diagrams (e.g., the round function of iShadow) could be more detailed to aid understanding. The raw data is not provided, which is a significant omission given PeerJ's policy on data availability.

Experimental design

1. Originality and Scope: The paper presents original research within the scope of the journal, focusing on a lightweight AE algorithm for IoT. However, the novelty of the proposed iAESR algorithm is not sufficiently justified. The enhancements to the Shadow algorithm (iShadow) are described, but the paper does not clearly articulate how these improvements significantly advance the state-of-the-art.
2. Research Question: The research question is relevant but not well-defined. The paper does not explicitly state how the proposed algorithm addresses a specific, identified gap in the current literature or practice.
3. Methodology: The methods are described with sufficient detail, but there are concerns about the rigor of the investigation. The security analysis, while thorough, lacks empirical validation. The paper relies heavily on theoretical proofs and does not provide sufficient experimental results to demonstrate the practical security and performance of the proposed algorithm.

Validity of the findings

1. Data Robustness: The paper does not provide raw data or sufficient experimental results to support its claims. The security analysis is based on theoretical assumptions, and there is no empirical evidence to validate the robustness of the proposed algorithm against real-world attacks.
2. Statistical Soundness: The paper lacks statistical analysis or experimental validation of the proposed algorithm's performance. While the authors claim that the algorithm is efficient and secure, these claims are not backed by concrete data or comparative analysis with existing algorithms.
3. Conclusions: The conclusions are well-stated but are limited by the lack of empirical evidence. The paper concludes that the iAESR algorithm is suitable for IoT environments, but this conclusion is not fully supported by the results presented.

Additional comments

The paper presents an interesting approach to lightweight authenticated encryption for IoT, but it falls short in several critical areas. The lack of empirical data and experimental validation significantly weakens the paper's claims. Additionally, the novelty of the proposed algorithm is not sufficiently justified, and the paper does not clearly articulate how it advances the state-of-the-art in lightweight cryptography.

Cite this review as

·

Basic reporting

The paper introduces a lightweight authenticated encryption (AE) for IoT using iShadow, an enhanced version of the Shadow block cipher, and employs duplex sponge construction for efficiency. The security analysis addresses essential aspects like differential and linear cryptanalysis but lacks formal security proofs (e.g., IND-CPA, IND-CCA).

Key points for improvement include:
1. Limited justification for iShadow enhancements, requiring a deeper comparative analysis and rigorous impact assessment of operations.
2. Weak justification for parameter selection in iAESR configurations (e.g., 16 rounds), without discussing their impact on security and efficiency.
3. Ambiguity in security claims related to Table 6's 64-bit and 128-bit security levels, needing empirical validation against attacks.
4. Performance evaluation is inconclusive without full hardware specifications and missing power consumption data, which is critical for IoT. Comparing software and hardware acceleration implementations is necessary.

Experimental design

The experimental methodology follows a structured approach, including implementation details and empirical performance evaluations. Performance results for different data sizes help assess scalability.
However, the following issues & weaknesses need to be addressed:
1. Missing Computational Complexity Analysis-
The paper does not explicitly analyze the computational complexity of encryption/decryption operations.
A complexity breakdown (e.g., O(n) notation for encryption, decryption, and authentication) would improve clarity.
2. Lack of Power Consumption Data
Since the paper targets IoT applications, it must include power efficiency results.
A comparison of energy consumption per encryption/decryption operation would validate real-world feasibility.
3. No Comparative Hardware Implementation
The paper claims iAESR is hardware-efficient, but there is no FPGA/ASIC implementation.
Including hardware benchmarking (e.g., gate count, memory usage, latency) would strengthen claims of lightweight performance.

Validity of the findings

The paper claims that iAESR is secure against differential, linear, and algebraic attacks, but no formal security proof is provided. Security should be demonstrated under standard cryptographic models like IND-CPA (Confidentiality), INT-PTXT (Integrity), or IND-CCA (Chosen Ciphertext Attack Resistance). Without formal proof or security reduction, the security claims remain unverified.

Unclear Justification of Security Margins:
Table 6 claims 64-bit and 128-bit security levels but does not justify how these values are derived. A security-bound analysis that considers attack complexities, brute-force resistance, and cryptanalysis results.

Ambiguous Improvement in iShadow Over Shadow:
The authors state that iShadow enhances security, but no empirical security comparisons are provided. A comparative differential analysis between Shadow and iShadow.
Performance benchmarks for diffusion, avalanche effect, and cryptanalysis resistance.

No Statistical Variance or Error Analysis in Performance Results:
Table 8 reports average execution times, but no standard deviations or confidence intervals are provided. A repeatability study with multiple test runs and error margins to show how consistent the results are is missing.

Additional comments

The paper presents a promising approach to lightweight AE for IoT; however, it requires stronger security proofs, deeper experimental validation, and improved formatting. Addressing these issues would significantly enhance its impact and credibility.

**Recommendations for Improvement:**

1. Provide a formal security proof to establish robustness under cryptographic security models.
2. Explain how iShadow improves diffusion and resistance against attacks, including quantitative comparisons.
3. Clarify the rationale for parameter choices, such as key size, number of rounds, and nonce length.
4. Include a power consumption analysis, as energy efficiency is critical for IoT applications.
5. Extend performance benchmarking to include hardware implementations, such as FPGA and ASIC.
6. Provide more experimental details, including CPU type, RAM usage, and clock cycles per operation.
7. Correct typographical errors and grammar inconsistencies. For example, line 67 contains a redundancy with "the the complexity." The term "diffusivity" in line 249 is uncommon; consider using "diffusion properties." Use "modulo addition" instead of "addition module," and address inconsistent punctuation. Some explanations are lengthy and should be broken into shorter, clearer statements.
8. Improve table formatting for better clarity.
9. Add error bars to performance graphs to indicate result variance.

Reviewer 3 ·

Basic reporting

1. While the abstract mentions the integration of iShadow with a duplex sponge structure, it lacks specific technical details about how this integration improves security or efficiency.

2. The abstract does not provide any quantitative results or performance metrics (e.g., reduced computational overhead, memory usage, or energy consumption). Including such details would strengthen the claim of the scheme’s efficiency and suitability for IoT devices.

3. The abstract states that the scheme ensures "security and integrity of communication" but does not elaborate on how this is achieved or what specific security properties (e.g., confidentiality, authenticity, resistance to attacks) are guaranteed.

4. The contribution section of the paper under the introductory section needs to be rewritten, summarized, and made clear. The author can improve readability by breaking long sentences into concise statements. For example:
- Improved the Shadow algorithm by adding a left rotation after each round (except the last) to strengthen security.
- Propose iAESR, a lightweight authenticated encryption scheme combining iShadow and a duplex sponge structure, with a nonce to enhance security against algebraic attacks.
- Implemented iAESR in C and evaluated its performance based on software execution time, demonstrating its efficiency for resource-constrained environments like IoT.

5. The conclusion section does not provide specific results or metrics to support the claim of iAESR’s superiority in efficiency and security. Including quantitative findings (e.g., execution time, memory usage, resistance to specific attacks) would strengthen the conclusion.

6. Also, the conclusion briefly mentions the components of iAESR (initialization, associated data processing, etc.) but does not explain how these components contribute to its security or efficiency. I think a bit more technical detail would make the conclusion more informative.

7. Some points, such as the lightweight characteristics of iAESR and its suitability for IoT, are repeated from earlier sections. I think the conclusion could focus more on synthesizing the findings and their implications rather than restating known information.

8. The future work section could be more specific by mentioning directions like hardware implementation, IoT protocol integration, or benchmarking against other lightweight AE algorithms.

Experimental design

Even though the author tries to describe the process of proposed algorithm very brief in the ISHADOW and Basic properties sections. However, there should be a Section "Methodology or Method" to describe details of the whole process for easy understanding.

Validity of the findings

-

Cite this review as

---

## Round 0.2 · accepted · Accept

The authors addressed all the concerns and the manuscript is accepted for publication.

Reviewer 1 ·

Basic reporting

It is up to my satisfaction.

Experimental design

It is up to my satisfaction.

Validity of the findings

It is up to my satisfaction.

Additional comments

none

Cite this review as